# Coronary Computed Tomography vs. Cardiac Magnetic Resonance Imaging in the Evaluation of Coronary Artery Disease

**DOI:** 10.3390/diagnostics13010125

**Published:** 2022-12-30

**Authors:** Lukas D. Weberling, Dirk Lossnitzer, Norbert Frey, Florian André

**Affiliations:** 1Department of Cardiology, Angiology and Pneumology, University Hospital Heidelberg, Im Neuenheimer Feld 410, 69120 Heidelberg, Germany; 2DZHK (German Centre for Cardiovascular Research), Partner Site Heidelberg/Mannheim, 69120 Heidelberg, Germany

**Keywords:** cCTA, CMR, cardiovascular imaging, coronary artery disease, stress test, CMR perfusion

## Abstract

Coronary artery disease (CAD) represents a widespread burden to both individual and public health, steadily rising across the globe. The current guidelines recommend non-invasive anatomical or functional testing prior to invasive procedures. Both coronary computed tomography angiography (cCTA) and stress cardiac magnetic resonance imaging (CMR) are appropriate imaging modalities, which are increasingly used in these patients. Both exhibit excellent safety profiles and high diagnostic accuracy. In the last decade, cCTA image quality has improved, radiation exposure has decreased and functional information such as CT-derived fractional flow reserve or perfusion can complement anatomic evaluation. CMR has become more robust and faster, and advances have been made in functional assessment and tissue characterization allowing for earlier and better risk stratification. This review compares both imaging modalities regarding their strengths and weaknesses in the assessment of CAD and aims to give physicians rationales to select the most appropriate modality for individual patients.

## 1. Introduction

Coronary artery disease (CAD) represents a widespread burden to both individual and public health, steadily rising across the globe [1]. Thus, safe and accurate tests for diagnosis and risk stratification are of paramount importance. In recent years, the European Society of Cardiology, American Heart Association and National Institute for Health and Care Excellence (NICE) updated their guidelines on the diagnosis and management of chronic coronary syndromes and in particular heightened the importance of non-invasive anatomical or functional testing in patients with suspected CAD [2,3,4]. Coronary computed tomography angiography (cCTA) is the imaging of choice for anatomical testing in most patients with suspected CAD. Cardiac magnetic resonance imaging (CMR) is widely recognized as an accurate, well-validated, non-ionizing imaging technique [5]. In comparison to other functional testing modalities, it uniquely incorporates a high diagnostic accuracy with the possibility to assess cardiac morphology, function, and tissue composition. This is why we chose it as representing modality to compare with cCTA, although many arguments may also hold true for other functional testing modalities such as nuclear imaging. Due to its high value in the recent guidelines, usage of both cCTA and stress CMR doubled in the last decade [6]. In this review, we compare the strengths and limitations of both imaging modalities and provide guidance on how to select the best suitable modality for each patient.

## 2. Diagnostic Performance of cCTA vs. CMR

Both cCTA and vasodilator stress CMR are robust imaging modalities with reported diagnostic image quality in >97% of cases [7,8].

However, to compare both in their diagnostic capabilities, comprehension of what both modalities primarily test for is indispensable. In general, cCTA is widely used for the assessment of the anatomical presence of coronary artery disease, whereas stress CMR tests for the functional presence of ischemia.

cCTA has the highest sensitivity (97%) of all non-invasive imaging modalities for the detection of anatomically significant CAD, but low specificity for anatomically (78%) or functionally (53%) significant CAD [9,10]. 

The latter is often due to partial volume or blooming artifacts which can lead to an enlargement of calcified plaques, thus resulting in an overestimation of stenosis severity [11]. Consequently, a high grade of calcification, as measured with the Agatston score, is the most relevant predictor of an uninterpretable cCTA. The same artifacts impair the assessment of coronary artery stents.

However, cCTA allows for the identification of patients in the early stage of CAD having non-obstructive coronary plaques. These patients have long been underrecognized since these plaques typically do not cause reduced blood flow and consequently most non-invasive tests assessing ischemia are inconspicuous. Yet, these patients are at increased risk since the rupture of non-obstructive plaques is the main cause of myocardial infarction [12,13]. Furthermore, cCTA is able to assess the overall plaque burden and to detect signs of plaque vulnerability such as low-attenuation, spotty calcifications, low-attenuation plaque, and the so-called “Napkin ring sign” (a plaque with a low attenuating necrotic core surrounded by a thin ring-like hyperattenuating rim) [14,15].

The identification of additional patients at risk by cCTA improves patient outcome significantly, as shown in the PROMISE, CONFIRM and SCOT-HEART trial [16,17,18,19]. For the PROMISE trial, 9,102 patients with intermediate pretest probability for obstructive CAD were randomly assigned to functional testing or anatomical testing with cCTA [17]. The primary endpoint of a major adverse cardiac event (MACE) was not statistically significant between groups, but cCTA was associated with fewer invasive angiographies showing no obstructive CAD [20]. In a post hoc analysis a normal cCTA, in contrast to a completely normal functional test, was highly unlikely to be associated with a MACE for at least 2 years and the authors attribute this effect to the identification of patients with non-obstructive CAD [17]. However, over-generalization of this study with regard to an inferiority of functional testing should be avoided, since patients had mostly atypical angina (77.8%), the Framingham risk score was intermediate or high in 77.1% and mostly nuclear perfusion stress (67.8%) and no stress CMR was used [17]. Additionally, 10% of patients had undergone exercise treadmill testing, which has low diagnostic accuracy [2]. Nevertheless, the PROMISE trial provides insights into how patient outcome might be improved apart from identifying ischemia alone.

In the SCOT-HEART trial, 4,146 patients with angina and suspected CAD were randomly assigned to standard care or standard care and cCTA [18]. In the cCTA group occurrences of fatal and nonfatal myocardial infarctions were halved as opposed to standard care [19]. The additional cCTA changed treatment significantly, with more appropriate use of invasive angiography, more preventive treatment, and lesser antianginal treatment [18,19]. Overall, 1 in 4 patients had changes to their treatment [18]. In the CONFIRM trial assessing 27,125 patients, statin use but not acetylsalicylic acid (ASA) use, was associated with a significant reduction in mortality for individuals with non-obstructive CAD, but not for individuals without CAD [16].

The same was observed by a registry study on 33,552 patients with <50% coronary stenoses on a cCTA, in which a new medication with a statin after cCTA was associated with a reduction in myocardial infarctions and all-cause death, directionally proportional to the observed CAD burden [21]. cCTA is, therefore, able to identify those patients in which LDL-lowering therapy is most beneficial and has been shown to significantly influence prescription rates of statins in vulnerable patients [22,23]. 

Apart from coronary stenosis assessment and plaque characterization, cCTA also allows for visualization and quantification of perivascular fat suggestive of coronary inflammation, which is shown to be a negative predictor for cardiac outcome [24,25].

In summary, cCTA is the diagnostic modality of choice to safely assess the presence of CAD, independently of its hemodynamic relevance, and, thus, guide further treatment such as the initiation of preventive medical therapies. cCTA can achieve non-inferior diagnostic accuracy in stable CAD patients without having the periprocedural risks of an invasive coronary angiography [26]. 

Stress CMR cannot safely exclude the presence of CAD. Its strength lies in the accurate assessment of the hemodynamic relevance of coronary stenosis, i.e., the testing for ischemia. Ischemia, however, is not a bivalent marker, but rather a continuum [27]. CAD with hemodynamically significant coronary artery stenosis leads to a decreased blood perfusion of the heart, which is dependent on the severity. During physiological or medication-induced stress, cell metabolism, diastolic function and systolic function are impaired, leading to ECG changes and ultimately symptoms such as angina or dyspnea [27]. As opposed to stress echocardiography, which usually screens for systolic wall motion deficits occurring relatively late in the ischemic cascade, perfusion stress CMR visualizes the decreased blood perfusion occurring early in the ischemic cascade [27]. In stress perfusion CMR, a vasodilator is applied, resulting in reduced myocardial perfusion in post-stenotic segments through a “steal phenomenon” (meaning that blood is predominantly directed through non-obstructed coronary arteries) and loss of autoregulation mechanisms, which is visualized by the application of a contrast agent [5,28]. Figure 1 shows an example of a perfusion deficit in perfusion stress CMR. Perfusion stress CMR has been shown to be non-inferior to invasive fractional flow reserve (FFR) with respect to major adverse cardiac events, to predict patient outcomes and to significantly reduce the need for both diagnostic and therapeutic invasive coronary angiography, thus guiding therapeutic decision making [29,30,31,32]. Apart from perfusion testing, CMR also offers an evaluation of systolic function and scar. Ejection fraction as measured by CMR is the current reference standard for systolic function due to its very low inter- and intra-rater variability (<3%) [33]. Late Gadolinium Enhancement (LGE) on the other hand is the reference standard for visualization of myocardial scar or tissue fibrosis. It may be used for targeted ablation in patients with atrial or ventricular arrhythmias or prediction of viability before revascularization [34,35,36]. Both have a significant correlation to patient outcome in a variety of entities and serve as early risk factors for adverse events [33,37,38,39,40]. The variety of sequences for tissue characterization as offered by CMR is of unparalleled importance in entities such as myocardial infarction with nonobstructive coronary arteries (MINOCA), myocarditis, storage diseases or cardiomyopathies [41,42,43].

In summary, CMR is the modality of choice to assess hemodynamic relevance, predict prognosis and for a thorough differential diagnosis in suspected CAD patients. 

## 3. Advances in cCTA

Broad advances have been made in the technical capabilities of contemporary CT scanners. High-end CT detectors with ≥128 slices have become a standard in many academic and non-academic sites in developed countries [7].

While conventional CT utilized a single polychromatic X-ray beam received by a single detector, dual-energy CTs (DECT) show increasing popularity. Depending on the vendor, different techniques are used. One is a second tube and detector unit at a 90° angle (dual *source* CT when operated with two different energies). An alternative technology is the combination of a single x-ray source rapidly alternating between low and high energies (fast switching) with a single detector that registers information from both energies [44]. Another technology is a detector made of two layers (sandwich detector) simultaneously detecting two energy levels [44]. DECT offers the possibility for spectral imaging that can improve tissue characterization, aid in improved composition analysis of coronary plaques and may reduce artifacts and contrast agent dose. Each technique has its own strengths and limitations. There are no reports comparing radiation dose among vendors [44]. 

Dual source scanners stand out with improved temporal solution (when both sources emit identical energy levels) and therefore a reduction in motion artifacts, which is critical in the assessment of a moving structure such as the heart [45]. They can also be used to enable ultra-high pitch spirals to reduce radiation exposure.

Regarding unenhanced CT such as coronary calcium scoring, the application of a tin filter may decrease the radiation dose in unenhanced scans without negatively affecting subjective image quality [46]. The tin filter reduces the proportion of low-energy photons and therefore increases the average photon energy. 

The most recent step in hardware advances for cCTA is the so-called photon-counting CT. It uses photon-counting detectors that separately register the energy of each photon. These offer a smaller pixel size and do not require the coating of each detection pixel by an optical reflector, which accounts for a 2- to 3-fold higher resolution than conventional CT (approximately 250 µm). In a recent clinical study photon-counting coronary CT led to significantly improved image quality (detectability indexes 2.3-fold to 2.9-fold higher) at a comparable radiation dose in 14 patients who underwent both standard energy-integrating detector dual-layer cCTA and photon-counting CT [47]. These results are promising since the higher resolution allowed for the visualization of smaller coronary vessels and the improved image quality was most evident in the presence of coronary stents and calcifications, which both are associated with impaired diagnostic image quality in standard cCTA. An example of a photon-counting cCTA scan is provided in Figure 2.

Apart from hardware improvements, computational advances with iterative or model-based reconstruction algorithms have improved image quality and reduced radiation dose [48,49,50]. 

Since invasive FFR is the current reference standard to assess the hemodynamic relevance of coronary stenosis, great effort has been directed into implementing a non-invasive cCTA-derived FFR value that can be interpreted similarly to its invasive equivalent [2,51]. 

Different techniques have evolved that are either applied in a core lab or on-site. Models based on fluid dynamics are computationally demanding, whereas recent advances in machine learning enabled faster on-site calculation [14,52,53]. The analysis is performed as post-processing using standard clinical cCTA images and does not need additional radiation. 

A number of studies show the discriminatory power of CT-FFR for the prediction of hemodynamic relevance to be superior to cCTA alone when compared to invasive FFR or stress CMR, improving diagnostic accuracy and especially specificity [51,54,55,56]. Implemented in the diagnostic work-up of patients with suspected CAD, it was able to significantly lower the rate of invasive angiography showing no obstructive CAD when compared to cCTA alone [57]. This was achieved without a negative impact on clinical outcomes [57]. Of note, machine learning-derived risk scores have been shown to add additional value to the prediction of inducible ischemia [58]. cCTA with CT-FFR has also been shown to have a high agreement with the decision derived from invasive coronary angiography in patients with left main or three-vessel CAD being evaluated for coronary artery bypass surgery and may also be combined with cCTA-derived features of plaque vulnerability providing an even better prognostic stratification [59,60]. Figure 3 gives an imaging example of CT-FFR.

Another example of cCTA exceeding the limits of pure anatomical testing for CAD is CT perfusion (CTP). The anatomical visualization of coronary arteries by cCTA is followed by vasodilator stress and repeated CT imaging to assess perfusion deficits in patients where cCTA alone was not able to exclude hemodynamically relevant stenosis [61]. CTP may also be performed without vasodilator stress, although the diagnostic value is limited [62].CTP has been shown to possess incremental diagnostic value over cCTA alone and was comparable to PET-CT or vasodilator stress CMR with respect to invasive FFR as reference [61,63,64,65,66,67]. An example is given in Figure 4. Limitations include the increased radiation exposure, longer and unpredictable scanning times since CTP only is used when the on-site imaging specialist is not able to exclude hemodynamically relevant stenosis on the first images, and the current lack of large multi-center and multi-vendor studies [61].

## 4. Advances in CMR 

Extensive advances have also been made in the field of CMR introducing a more robust imaging acquisition, new acquisition sequences, and vast post-processing possibilities. The relatively long acquisition times remain a modality-dependent issue but decreased considerably. For example, acquisition time of cine images was reduced by ~80% through the introduction of compressed sensing [68]. In addition, fast and native T1, extracellular volume and T2 mapping-derived parameters have become a promising biomarker for ischemic and non-ischemic cardiomyopathies, potentially reducing the need for time-consuming LGE imaging for myocardial tissue characterization [69,70,71]. Furthermore, developments in machine learning may render CMR scans without time-consuming ECG-triggering and breathing maneuvers possible, although these techniques have not been integrated into clinical routine yet [72,73].

The introduction of myocardial strain measurements boosted the assessment of myocardial function with superior predictive power than ejection fraction [74]. Myocardial strain measures the deformation of the myocardium during the cardiac cycle and has been shown to be an early risk marker in a wide spectrum of cardiac diseases such as of cardiotoxicity in cancer patients or heart failure in asymptomatic patients [75,76,77]. In patients following acute cardiac events such as myocardial infarction, reduced strain-derived parameters were associated with future cardiac events [78]. Strain measurements can be acquired without the use of contrast agents, fast (down to in one heartbeat) and with low inter- and intra-observer variability [79,80]. 

Improved resolution achieved by combining undersampling and motion correction has also allowed obtaining sub-millimeter isotropic CMR images. In proof-of-concept studies, the derived coronary lumenography visualizes coronaries at a quality that is sufficient to screen for coronary anomalies and even (to some degree) stenoses [81,82].

In stress CMR, the introduction of quantitative rather than qualitative perfusion imaging has decreased reader dependency and enabled a faster and simpler analysis [83]. Furthermore, it allows for the identification of globally reduced blood flow [83]. As an alternative to perfusion CMR, some proof-of-concept studies have shown a promising diagnostic value of medication-free protocols using, e.g., hyperventilation or a dynamic handgrip exercise as a stressor [84,85]. These can be combined with a number of advanced CMR sequences such as myocardial strain or myocardial oxygenation to create a completely medication- and needle-free CMR exam [84,86,87,88]. However, these studies currently lack large multicenter trials and have therefore not been introduced into clinical routine yet.

## 5. Safety Profile of cCTA vs. CMR

Risks of both cCTA or stress CMR are linked to the technique itself, the application of contrast agent or necessary stress medication in the course of the exam.

CMR is a non-ionizing technique working with a static magnetic field and radiofrequency pulses that may be causative for (reversible) headaches or vertigo [89]. Besides, a careful patient selection and preparation by experienced personnel is necessary to avoid interactions of metallic items (e.g., piercings, medical implants) with the electromagnetic field [90]. cCTA employs ionizing radiation coming along with stochastic health effects and potential damage to cells and genetic material [91]. However, technical advances have led to an impressive decrease in radiation dose. In the PROTECTION VI study including 61 hospitals in 32 countries, the radiation dose decreased by 78% between 2007 and 2017 without an increase in non-interpretable exams [92]. Foldyna et al. recently reported an effective radiation dose of 4.5 mSv for cCTAs in a large multi-center study of 64,317 performed scans in both academic and non-academic sites [7]. This must be seen in context to the average yearly background radiation of approximately ~3.1 mSv and would roughly translate to an excess cancer risk of <1 in 1000 [93,94]. Nevertheless, the large variability of the radiation dose between sites (up to 37-fold) remains problematic and underlines the importance of trained personnel and modern equipment [92].

Acute adverse reactions can occur after both CT and CMR contrast agent application, although slightly more frequently in Iodine-based CT (0.2–0.4%) than Gadolinium-based MRI (0.1–0.18%) contrast agents [7,95,96,97]. Specific adverse effects of Iodine-based CT contrast agents include kidney injury (~2.6%) and disturbance of thyroid function [98]. Gadolinium-based MRI contrast agents have been linked to possible tissue deposition (e.g., the brain) and are in very rare cases causative for nephrogenic systemic fibrosis in patients with severely impaired renal function [99,100]. However, tissue deposition is currently of unknown clinical significance and both entities occur significantly rarer (in most studies to the limits of non-existence) with the use of modern macrocyclic contrast agents as opposed to previously used linear contrast agents [100,101,102,103,104,105].

Apart from contrast agents, both cCTA and stress CMR usually require specific medication. For cCTA, beta blockers and glyceryl trinitrate (GTN) are used to achieve adequate heart rate control, if necessary, and to dilate coronary arteries prior to the exam [97]. These may lead to blood pressure drops, which is why there should be no application of GTN if systolic blood pressure is <90 mmHg [97]. Generally, both medications are well tolerated in outpatients as well as in inpatients with rare occurrences of vasovagal symptoms (~0.3%) or cardiac arrhythmias (~0.1%) [7,97]. In contrast, caution is advised in non-stable patients and patients with aortic stenosis.

For CMR, vasodilator stress CMR with regadenoson, dipyridamole, or adenosine is the current standard for patients with suspected CAD. Of those, regadenoson is best tolerated due to its high specificity to the A_2A_ receptor [8]. Common adverse events include a paroxysmal AV block (<0.3%), hypotension (<0.2%), angina (<0.1%) and bronchospasms (0.5–0.8%) [8]. Patients with a history of COPD and especially asthma are at higher risk for bronchospasm, which is why the use of regadenoson should be considered in them [8]. Although the incidence of non-fatal adverse events is higher in patients with pre-existing cardiac conditions, serious adverse events still remain low [106]. As an alternative to perfusion CMR using vasodilator stress, dobutamine stress CMR analyzing ischemia-induced wall motion abnormalities can be conducted. However, higher rates of adverse events (i.e., non-sustained ventricular arrhythmias in 0.4%, atrial fibrillation in 1.6%), patient discomfort (i.e., severe chest pain or dyspnea in 4.0%) and longer exam times have limited its usage in clinical routine [107]. Nevertheless, it may be advantageous in advanced CAD cases with previous coronary artery bypass graft surgery, large myocardial scars with unknown viability, or total coronary artery occlusion in which stress perfusion is often inconclusive.

In summary, both cCTA and vasodilator stress CMR exhibit excellent safety profiles.

## 6. Discussion

Over the last decade, cCTA has evolved from an imaging modality, which was mainly used to exclude significant CAD in patients at a rather low risk, to a valuable method for risk stratification and therapeutic decision making resulting in a significant improvement in patient outcome. Image quality has improved, radiation exposure has decreased, and anatomical information can be complemented by information on the hemodynamic relevance of stenoses. Consequently, it is recommended as a first-line imaging modality in patients with stable chest pain in the current guidelines [3,4,108]. Hence, it may seem like the undisputable imaging modality in suspected and proven CAD patients.

However, several factors limit the use of cCTA and the most relevant are summarized in Table 1. First, the risk of Iodine contrast-induced nephropathy has to be taken into account in patients with impaired renal function, a common comorbidity in cardiac patients [98]. This and the associated radiation exposure make cCTA unappealing for frequent follow-up examinations. Furthermore, patients with heart rhythm disorders (e.g., premature contractions, atrial fibrillation) still reveal robust CMR stress perfusion imaging, whereas cCTA image quality might be impaired and possibly necessary acquisition repetitions come at the cost of increased radiation and contrast agent exposures.

Second, although assessment of coronary stents has improved, assessment of in-stent stenosis and their functional assessment with CT-FFR is hampered and in some cases even not possible. In addition, the evaluation of patients with severe and extensive coronary calcifications, coronary artery occlusion (with collateral circulation) or previous coronary artery bypass graft surgery can be challenging. 

In CT-FFR negative stenoses, additional factors such as symptom burden, current medication and comorbidities must be considered to decide whether further testing might be necessary. Here, the cCTAs and CT-FFRs blind spot concerning the presence of the coronary microvascular disease has to be mentioned [109]. CTP might be possible in those patients but has only recently been evaluated in a proof-of-concept study [110]. 

CAD evaluation by stress CMR is an important imaging modality in the diagnosis and risk stratification and, thus, is represented in current guidelines [2,3,108].

Of note, the concept of inducible ischemia is not yet fully understood and is the subject of current discussions [111,112]. The ISCHEMIA trial even questioned the prognostic impact of invasive revascularization at all [112]. However, considerable limitations, such as a significant number of revascularizations (25.7%) in the conservative arm and the inclusion of patients with no or mild ischemia (14.1%) or functional tests with low levels of evidence (25.0%) limit the validity of that conclusion [112]. Interestingly, studies comparing myocardial perfusion defects with myocardial oxygenation defects have shown those not to be identical in patients with CAD [113,114]. Nevertheless, patients without ischemia in stress CMR have a good prognosis demonstrating its value for risk stratification and therapeutic management [29,30].

In patients with myocardial scarring, the viability of ischemic regions and therefore the potential benefit of revascularization is important. In these patients, the diagnostic assessment goes beyond stenosis evaluation alone and CMR can provide information on both, ischemia and viability, in a single examination. Its ability to characterize myocardial tissue allows for differential diagnostics in patients with MINOCA as CMR can identify various etiologies such as myocarditis, tako-tsubo syndrome, and thromboembolic myocardial infarction. The visualization/quantification of arrhythmogenic substrate of scar and exact calculation of left-ventricular ejection fraction with CMR is also useful in clinical practice to decide for or against an implantable cardioverter-defibrillator in patients with ischemic and non-ischemic heart disease [108].

Regarding the choice for the appropriate imaging modality aside from the individual patient site-specific conditions should be taken into consideration. Especially, the diagnostic quality of cCTA depends on the scanner model. On a global scale, ≤64 slice CT detectors are still widely used and the possibilities for CTP or CT-FFR are limited in most areas due to technical and reimbursement restrictions [7]. As mentioned above, radiation dose also varies widely from site to site. On the contrary, the impact of the latest available scanner is less relevant in vasodilator stress CMR. On a global scale, non-imaging stress tests still play an important role especially in emerging or developing countries due to the low costs [115].

Apart from hardware, local staff expertise may also be different as not all sites offer cCTA and CMR. Reimbursement is different in every country and influences waiting times, local expertise, and referral choice [116]. The total costs of both modalities are not easy to compare since they depend on available hardware, software and staff in comparison to the occupancy rate of the scanner. Overall, stress CMR is more expensive than cCTA, but the possible recompensation is also higher. However, adjusted to a benefit in quality of life, CMR seems to be more cost-effective than cCTA [117]. 

The strength of cCTA evidently lies in the identification of additional at-risk patients, in which preventive approaches (e.g., lifestyle changes, medication) are beneficial. Here, the fact that >50% of all myocardial infarctions are caused by non-obstructive plaques has to be emphasized, so the initiation of preventive treatment is key in influencing patient outcome [17,118]. However, the benefit of cCTA diminishes without therapeutic consequences. In a patient already on optimal ASA and statin medication (e.g., for peripheral artery disease), no therapeutic benefit of early CAD detection can be achieved, if there is no need for revascularization. As comorbidities and cardiovascular risk factors become increasingly common in advanced age, an age-specific approach to the PROMISE trial showed anatomic testing to provide better prognostic discrimination for patients < 65 years, whereas functional testing offered better prognostic discrimination in patients > 65 years. 

The potential to detect early stages of CAD makes cCTA the diagnostic modality of choice for a large proportion of patients with suspected CAD, taking into account that 75% of patients in the SCOT-HEART trial had no obstructive CAD, despite a high prevalence of cardiovascular risk factors [18]. 

Although this review focuses on CAD patients, cardiac and pulmonary comorbidities also play an important role in the selection of the appropriate modality. While heavy smokers suffering from dyspnea may benefit from a combined assessment of the coronary arteries and the lungs, younger patients with a differential diagnosis of myocarditis or cardiomyopathy may benefit from myocardial tissue characterization offered by a CMR. 

In general, for CAD evaluation, cCTA is favorable in patients that are younger, have few comorbidities, no known cardiac disease and are currently not taking preventive medication, whereas stress CMR is more appropriate in older patients, in those with known CAD, already taking ASA/statin and in need of regular follow-ups. 

Of note, both cCTA and stress CMR can be used as a downstream diagnostic testing method in patients with an inconclusive test result of the primarily chosen modality.

Thus, it is important to view cCTA and stress CMR as mostly complementary diagnostic modalities, each possessing strengths and limitations but both providing valuable and accurate diagnostic information to guide physicians’ treatment decisions. Physicians should strive for an individual approach in each patient with regard to patient-related factors and patient wishes considering each modality’s strengths and limitations.

## Figures and Tables

**Figure 1 diagnostics-13-00125-f001:**
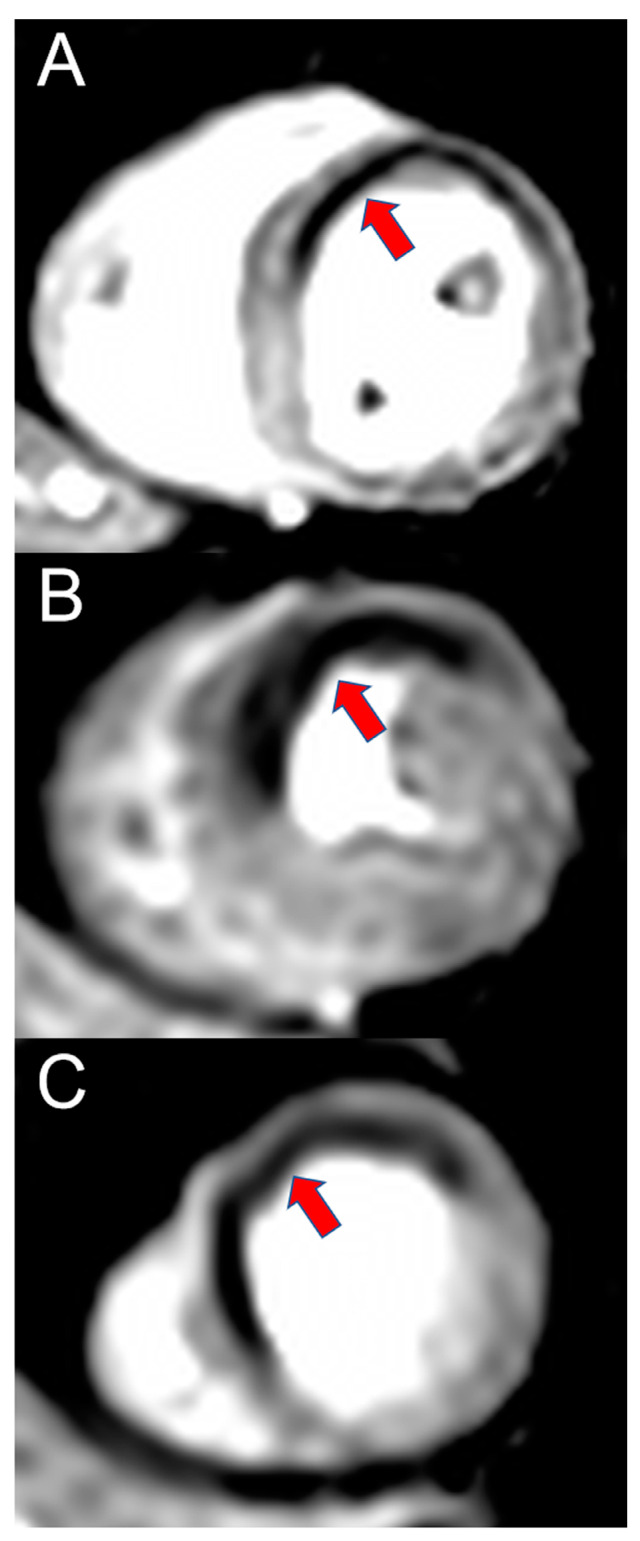
Imaging example of a stress CMR. A 58-year-old female patient with atypical angina symptoms and low pretest probability was referred for a stress CMR. The stress perfusion imaging revealed an extensive perfusion deficit (red arrows) of the anterior and anteroseptal wall on the basal (**A**), mid-ventricular (**B**) and apical (**C**) slice. An invasive coronary angiography showed a subtotal stenosis of the proximal left anterior descending artery (LAD) and the diagonal branch.

**Figure 2 diagnostics-13-00125-f002:**
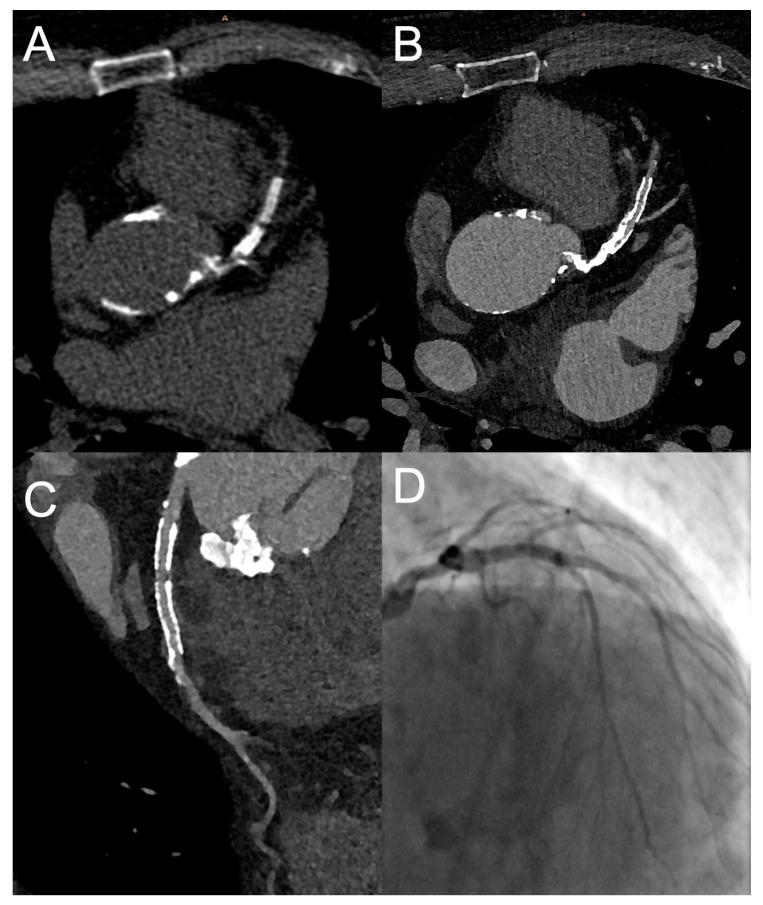
ECG-gated ultra-high-resolution cCTA of the heart using photon-counting. Images show the heart of a 79-year-old male patient referred for preprocedural planning of a transcatheter aortic valve replacement. Despite extensive calcifications (Agatston score of 3388) and two coronary stents in the LAD, a visualization of the coronary arteries was possible. (**A**): unenhanced coronary calcium scoring CT, axial reconstructions with a slice thickness of 2.0 mm using the Br36 kernel. (**B**): ultra-high-resolution cCTA, axial reconstructions with a slice thickness of 0.4 mm and an increment of 0.2 mm using the kernel Bv56 and moderate iteration Q3. (**C**): Curved multi-planar reconstruction of the LAD. An in-stent stenosis can be ruled out. (**D**): Invasive coronary angiography of the LAD confirming the CT findings. Courtesy of Christopher L. Schlett, University of Freiburg, Germany.

**Figure 3 diagnostics-13-00125-f003:**
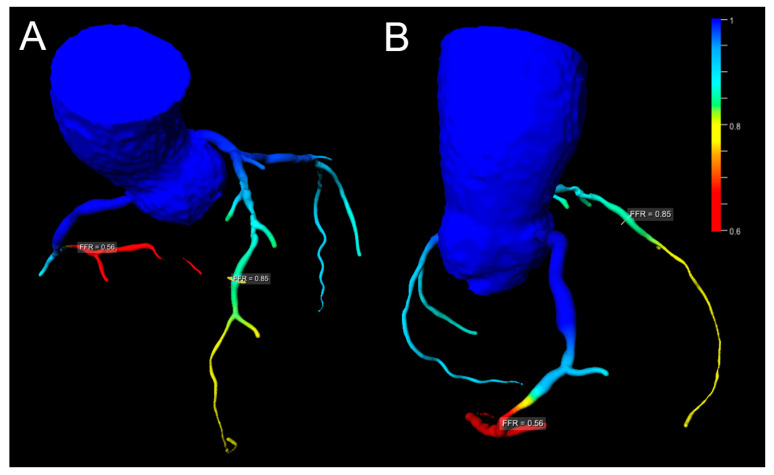
Imaging of a cCTA with non-invasive CT-FFR measurements. 65-year-old male patient referred for suspected CAD. Three-dimensional reconstruction of the coronary tree with color-coded FFR calculations. These show a hemodynamically relevant stenosis in the distal right coronary artery ((**B**), FFR 0.56), whereas stenosis in the distal LAD (**A**) is not hemodynamically relevant (FFR 0.85). Courtesy of Sebastian J. Buss, MVZ-DRZ Heidelberg, Germany.

**Figure 4 diagnostics-13-00125-f004:**
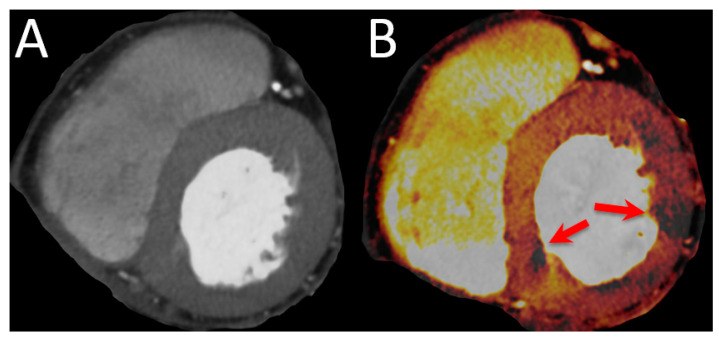
Imaging example of a CTP. A 78-year-old female patient with atypical angina and intermediate pretest probability. (**A**) shows the multiplanar reconstructions of the basal short axis with a DECT at 90 kV and 150 kV. (**B**) shows the perfused blood volume which is reduced in two segments (red arrows).

**Table 1 diagnostics-13-00125-t001:** Considerable patient factors to select either cCTA or stress CMR as primary diagnostic modality. The combination of the different factors should be taken into consideration to attain the best individual approach.

	Favor cCTA	Favor Stress CMR
CAD	Unknown	Known
Previous coronary stent	No	Yes
Preventive medical therapy	None	ASA, statin
Patient age	Middle-aged	Advanced, Young with known coronary anomalies
Pretest probability of CAD	Low/Intermediate	High
Regular follow-up needed	No	Yes
Previous diagnostic work-up	Inconclusive stress test	cCTA with non-diagnostic image quality or stenosis of indetermined hemodynamic significance
Metallic implants	Non-removable metallic implants without or with unknown MR safety, Non-removable metallic implants which may impair CMR image quality severely	-
Comorbidities	COPD/Asthma Severe claustrophobia	Hyperthyroidism, moderately/ severely impaired kidney function, high heart rate
Potential differential diagnosis other than CAD	Pulmonary or aortic pathology	Myocarditis, pericarditis, thrombembolism–MINOCA assessment
Viability assessment required	No	Yes
Assessment of myocardial edema, function, scar tissue or fibrosis required	No	Yes
Severe or extensive coronary calcifications expected or proven	No	Yes
Allergies	Allergy to Gadolinium-based contrast agents	Allergy to Iodine-based contrast agents
Claustrophobic	Yes	No
Modern CT scanners available (≥128 slice detectors)	Yes	No
Additional factors to be considered	Local expertise Timely availability (if necessary) Patient preference	

## Data Availability

No new data were created or analyzed in this study. Data sharing is not applicable to this article.

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
