# Peer review of "Coronary Computed Tomography vs. Cardiac Magnetic Resonance Imaging in the Evaluation of Coronary Artery Disease"

_diagnostics, 2022, doi:10.3390/diagnostics13010125_

Round 1

Reviewer 1 Report

In this review paper by D. Weberling et al., the authors cover two separate imaging modalities - cCTA and CMR for detecting and managing coronary artery disease and coronary ischemia. 

The paper is well organized and well written, showing the strengths and limitations of every modality. There are many concerns and many debates regarding the optimal imaging modality for each patient and for every clinical decision process, and this review nicely compares the two modalities. 

I have several general remarks. 

1. First, although cCTA is unique in it's ability to show coronary anatomy, plaque distribution and plaque burden, there are many modalities that can help assess inducible ischemia - from simple stress tests, to SPECT, PET perfusion and stress echocardiography. The authors should mention all the alternative modalities, explain the reason to choose CMR over any of them.

2. I think that the second paragraph, regarding safety of the various modalities should be somewhere near the final third of the paper (maybe before the discussion), and not where is currently appears. It can also be shortened.

3. The authors should comment about worldwide trends in the use of cCTA and CMR. Where is each modality more common, is the use increasing or decreasing? why?

4. At the discussion the authors briefly discuss the NICE criteria and the ESC guidelines for managing stable AP. The discussion should also include the American ACC\AHA guidelines and perhaps compare all of the different guidelines regarding the use of anatomical\physiological imaging.

5. There is no mention of the ISCHEMIA trial. I believe that this trial helped balance CTA and CMR tests. this review should show the balance between CT testing and ischemia testing as shown by this paper. 

Minor comments:

1. page 3, the Paragraphe on line 137 is hard to understand. Consider rephrasing.

2. is there a cost difference between the tests? 

Overall, I enjoyed reading this review and I think that it is publishable after minor revisions. 

Reviewer 2 Report

The review paper Coronary Computed Tomography vs. Cardiac Magnetic Resonance Imaging in the evaluation of coronary artery disease, by dr Weberling et al. presents the two techniques well and in a balanced matter. I find the review of interest and learned something; this is the best thing that can be said about any review. I am sure a number of colleagues will think the same.

I have two main criticisms, and a few minor criticisms (please see below). Firstly, the reason for the review is given by the sentence: “In this review, we compare strengths and limitations of both imaging modalities and provide guidance on how to select the best suitable modality for each patient.” Myself working on a daily basis with both cardiac MRI and coronary CT I full well understand the authors interest in these two developing techniques, but it is not well explained why these two techniques were chosen: Why was CMRI chosen whereas scintigraphy techniques/PET is not discussed? Again, I full well understand the importance of cardiac MRI, but the discussion on perfusion defects could full well include reference to PET studies?

Secondly, the authors do not discuss the recent developments of magnetic resonance imaging for coronary lumenography; this should be included. See fx Bustin et al., 2020 (ref given below).

Minor criticisms:

CMR may “cause head-aches or peripheral nerve stimulatin”. I have seen peripheral nerve stimulation at 7 Tesla (and perhaps a few instances at 3 T), but most cardiovascular CMR is performed at 1.5 Tesla where I have never seen this happen. Please provide a reference for these problems at 1.5 and 3 Tesla.

“Foldyna et al. 49 recently reported an effective radiation dose of 4,5 mSv for cCTA in a large …” please report how much this is in comparison with the back-ground raditiona and how much this would contribute to life time cancer risk. Such doses are difficult to interpret for people not usually working in the radiology department.

The authors write: “Patients with a history of COPD or asthma are at higher risk for brochospasm, which is why the use of regadenoson should be considered in them [20]”. Bronchospasm is very rarely seen in patients with emphysema/COPD and in practice little happens if patients do not have asthma with wheezing!?

called „Napkin ring sing“, it is probably “Napkin ring sign” (not “sing”), please explain the sign to the reader.

”9.102 patients” should be ”9,102 patients”. The same seems for most numbers of patients in studies.

Sometimes “cCTA” seems to  be abbreviated “CTA”, please go through the manuscript and correct as needed.

Please for the reader explain “a „steal phenomenon“ (ie that the blood is directed predominantly to non-obstucted coronary arteries).

References

Bustin A, Rashid I, Cruz G, et al. 3D whole-heart isotropic sub-millimeter resolution coronary magnetic resonance angiography with non-rigid motion-compensated PROST. J Cardiovasc Magn Reson. 2020;22(1):24. Published 2020 Apr 16. doi:10.1186/s12968-020-00611-5.
